# Particle Tracking Using Dynamic Water-Level Data

**Yuan Gao**

Department of Civil, Environmental, and Construction Engineering, University of Central Florida,
Orlando, FL 32816, USA; yuan.gao@Knights.ucf.edu; Tel.: +1-970-581-7124

**Abstract:** The movement of fluid particles about historic subsurface releases is often governed by dynamic subsurface water levels. Motivations for tracking the movement of fluid particles include tracking the fate of subsurface contaminants and resolving the fate of water stored in subsurface aquifers. This study provides a novel method for predicting the movement of subsurface particles relying on dynamic water-level data derived from continuously recording pressure transducers. At least three wells are needed to measure water levels which are used to determine the plain of the water table. Based on Darcy's law, particle flow pathlines at the study site are obtained using the slope of the water table. The results show that hydrologic conditions, e.g., seasonal transpiration and precipitation, influence local groundwater flow. The changes of water level in short periods caused by the hydrologic variations made the hydraulic gradient diversify considerably, thus altering the direction of groundwater flow. Although a range of groundwater flow direction and gradient with time can be observed by an initial review of water levels in rose charts, the net groundwater flow at all field sites is largely constant in one direction which is driven by the gradients with higher magnitude.

**Keywords:** particle tracking; groundwater; dynamic water level; gradient

## 1. Introduction

Groundwater serves as an essential source of public drinking water for municipalities and rural areas, agricultural irrigation, and industrial operations [1,2]. However, overuse of chemicals through industrialization in the past decades has led to widespread contamination of groundwater by a diverse suite of organic and inorganic compounds [3]. Although many contaminants can be naturally attenuated in the subsurface via microorganism activities, residual chemicals can persist for a long period, which has posed substantial harm to natural groundwater resources [4].

To mitigate the threats of subsurface contaminants to human health and the environment, efficient methods have been employed to track the flow of contaminants. Among those, particle tracking is commonly used to define the pathlines of solute particles under purely advective transport [5]. In particular, particle-tracking schemes have been formally incorporated into solute transport models to account for the advective component of transport [6]. The basic idea is to follow the movement of infinitely small imaginary particles placed in a flow field using either analytical or numerical methods [7]. Particle tracking has been widely used in numerical modeling of groundwater flow to track contaminant paths. For example, Cunningham et al. [8] described the information on the regional groundwater flow field as "inferred from particle pathlines". Maskey et al. [9] presented the use of different global optimization (GO) algorithms to determine the optimized combination of pumping rates and well locations for the removal of a contaminant plume using particle tracking. In a karst aquifer, a groundwater vulnerability zone was created using particle tracking based on the Head-Guided Zonation numerical method [10]. Alberti et al. [11] employed a Monte Carlo particle tracking method to assess the tetrachloroethene in groundwater and identified the diffusion area. Two modeling codes, MODFLOW and MODPATH [12], are commonly used for groundwater flow and particle tracking.

A backward particle-tracking method to delineate groundwater protection zones was reported as an effective and powerful tool in [13]. A new numerical technique called the convolution-based particle tracking (CBPT) method was developed to simulate resident or flux-averaged solute concentrations in groundwater models [14], which is valid for steady-state flow and linear transport processes such as sorption with a linear sorption isotherm and first-order decay. Yidana [15] also used particle tracking to define flow paths of the recharge in some aquifers in Ghana, and the particle tracking simulation identified travel times in the specific years from recharge areas to discharge areas along the flow paths. Moreover, cyclic water flow which may play an important role in groundwater flow also has been studied. The changes of water permeability characteristics could be influenced by cyclic water flow and corresponding water levels [16].

Although numerical models have been widely employed to track the movement of fluid particles, depending on spatial and temporal discretization, they may not be able to capture dynamic aspects of groundwater flow for complex water surfaces with dynamic water levels. Specifically, for complicated boundary conditions, numerical models are the first choice to resolve particle tracking, even though they contribute the proximate answers; however, compared to numerical models, particularly for some simple boundary conditions, temporal and spatial discretization in numerical simulations may be insufficient to accurately track particles. Analytical methods are useful techniques that can be applied to many ground water flow problems. This study explores a novel and precise analytical method to track particles using continuous water level data from a field site, providing an efficient tool for predicting the movement of subsurface contaminants by tracking particles. The objective of the study is to use continuous water level data to track particles thus serving an option to resolve groundwater flow under dynamic conditions. The focus of this study is the analytical method which is used to track particles. The geologic parameters used in this study are just to test the method and the values are assumed to be based on the local geologic conditions. Three geologic conditions where used for conducting particle tracking: (1) homogeneous, isotropic conditions, (2) homogeneous, anisotropic conditions with retardation, and (3) homogeneous, anisotropic conditions with degradation of contaminants in the subsurface following the first-order kinetics.

The paper is divided into four parts. The hydrogeology description of the study site is shown in Section 2. Computational methods are presented in Section 3. Results and a discussion are described in Sections 4 and 5. Finally, conclusions and recommendations for future work are presented in Section 6.

## 2. Description of the Study Site

The study site, the Pueblo Chemical Depot (PCD), Colorado, is located approximately 15 miles east of Pueblo, Colorado (Figure 1a). PCD was built to serve functions of ammunition, material storage, and a shipping center, and it was used to store chemical munitions in the late 1990s [17]. Sale et al. [17] reported that releases from PCD made plumes that were discharged into the Arkansas River alluvium. There are still several contaminants remaining in the subsurface at this site after source excavation, among which the hexahydro-1,3,5-trinitro-1,3,5-triazine (RDX) is the primary contaminant of concern. The groundwater table is found at about 2 to3 m below the surface. Sand is the main soil type above the water table. The site is underlain by 3–5 m of sandy alluvium. The alluvium is a fluvial terrace deposit associated with either the Arkansas River or Chico Creek. The alluvium is underlain by the Pierre Shale, which is hundreds of meters thick and extensive in area across the plains in south central Colorado [17]. The hydrogeologic cross section of the site is shown in Figure 2.

Daily frequency water level data were collected from five wells (Figure 1b) over the period from 1 March 2006 to 16 September 2008 (Figure 3). The data shown in Figure 3 are not continuous since the missing data were skipped in the study (29 August 2006 to 10 October 2006, 14 November 2007 to 22 April 2008). The variation in water level shown in Figure 3 basically followed the seasonal pattern: water level increased from September and maintained at a high level before starting to decline around June. Extreme hydrologic event, e.g., heavy storm or dry event, may contribute to the abnormal peaks during the general trend.

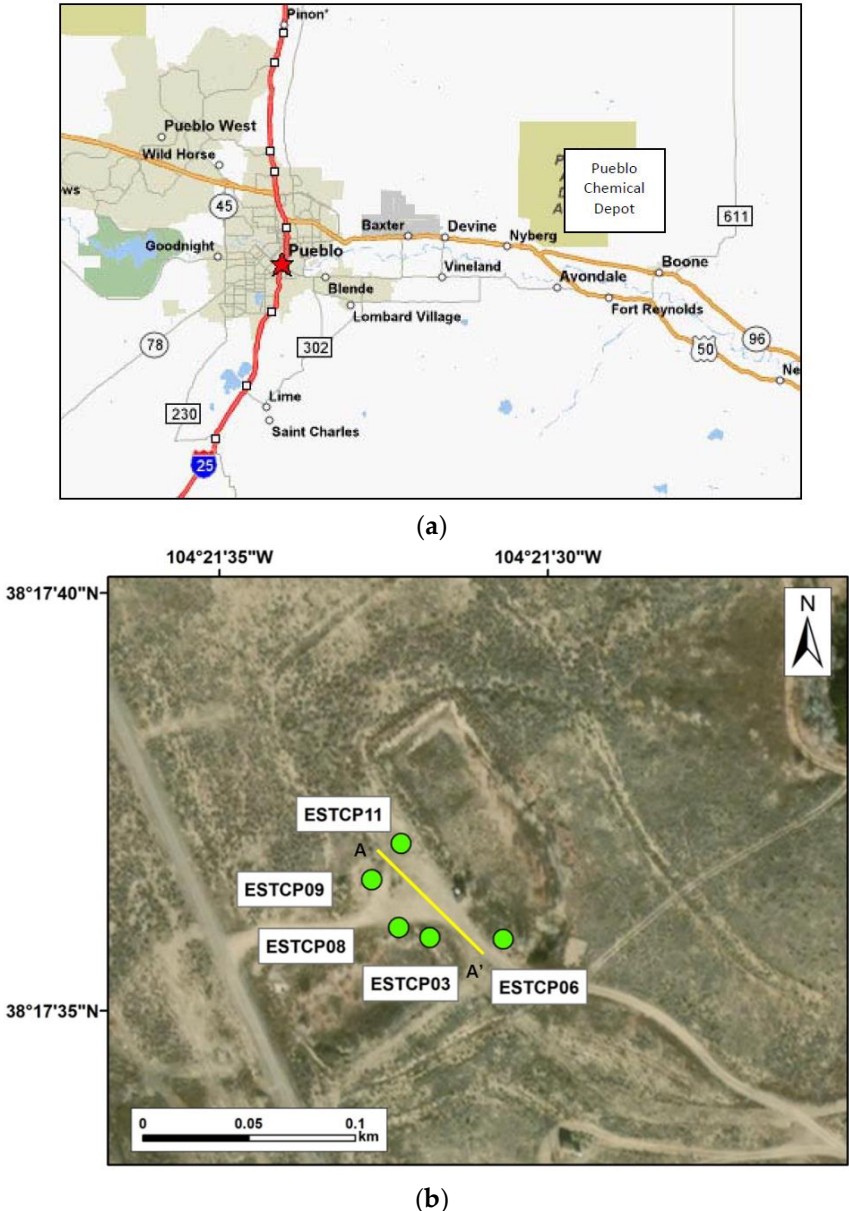

**Figure 1.** (**a**) Location of Pueblo Chemical Depot (PCD) near Pueblo, Colorado (CO) [17]. (**b**) Monitoring wells established in the Pueblo Chemical Depot area.

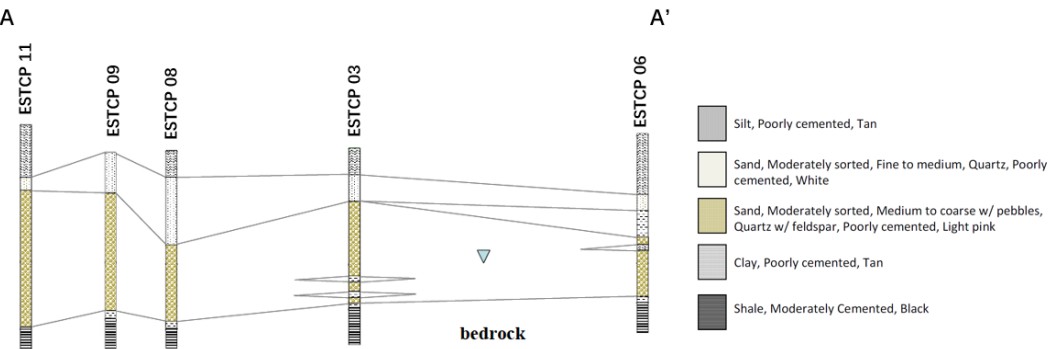

**Figure 2.** Geologic cross-sections through the demonstration location in Figure 1b [17].

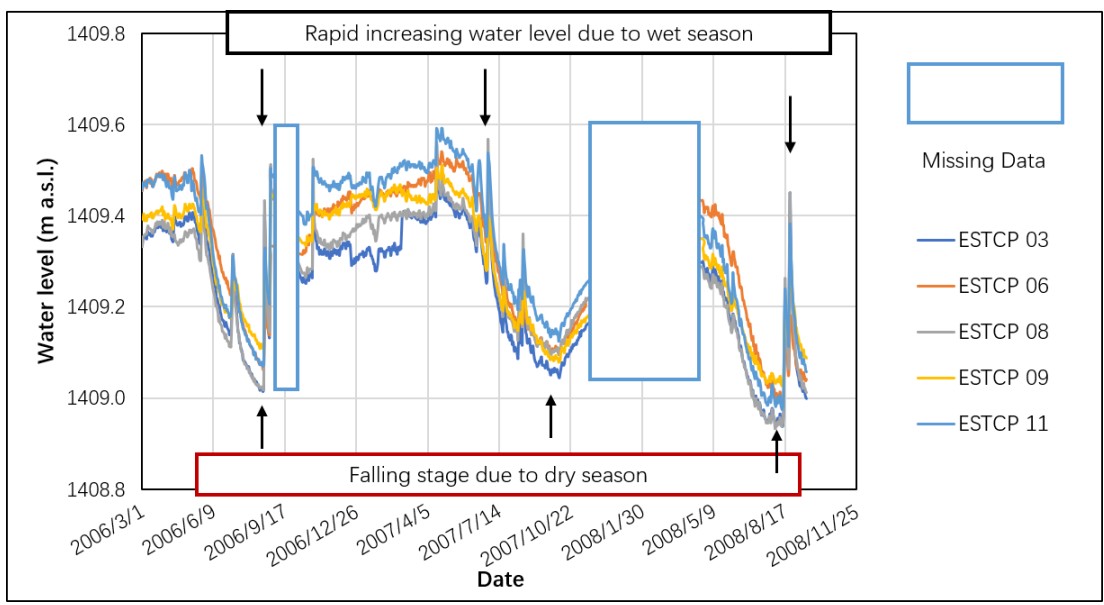

**Figure 3.** Water-level data in the five wells at Pueblo Chemical Depot (PCD), CO.

## 3. Methods

The basic idea for tracking fluid particles is: particles are placed in the system at an initial position, $x_0$, $y_0$, at an initial time, $t_0$. The position of the particles at any later time, t, is computed by solving the equations defined by the seepage velocity $v_x = q_x/\varphi = dx/dt$, and $v_y = q_y/\varphi = dy/dt$, where $\varphi$ is the effective porosity and $q_x$ and $q_y$ are Darcy velocity in the x and y direction, respectively [18]. The approach centers on using continuous-field water-level data obtained using pressure transducer data from monitoring wells. Three or more wells are used to resolve the plane of the groundwater surface below an area of interest at a prescribed time (Figure 4). Gradients in the $x$ and $y$ directions are employed in resolving the movement of fluid particles over a defined period of time. For each time step, the plane of the potentiometric surface is resolved, and transport vectors are added to one another, head to tail.

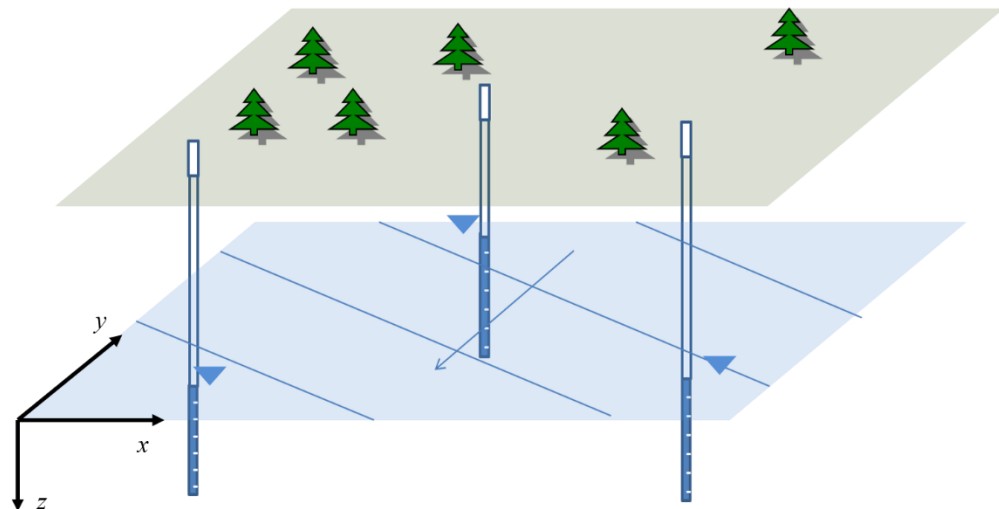

**Figure 4.** Three wells used to resolve the plane of the groundwater surface below an area of interest.

### 3.1. Application of Darcy's Law

Darcy's law is used to resolve the movement of a particle in $x$, $y$, and $z$ directions. Assuming a homogeneous and anisotropic material [19]:

$$q_x = -K_{xx}\frac{\partial h}{\partial x} - K_{xy}\frac{\partial h}{\partial y} - K_{xz}\frac{\partial h}{\partial z} \tag{1}$$

$$q_y = -K_{yx}\frac{\partial h}{\partial x} - K_{yy}\frac{\partial h}{\partial y} - K_{yz}\frac{\partial h}{\partial z} \tag{2}$$

$$q_z = -K_{zx}\frac{\partial h}{\partial x} - K_{zy}\frac{\partial h}{\partial y} - K_{zz}\frac{\partial h}{\partial z} \tag{3}$$

where, $q$ is the Darcy velocity ($LT^{-1}$) (L represents the length and T the time), $K$ is hydraulic conductivity ($LT^{-1}$), $h$ is hydraulic head (L), $x$, $y$, and $z$ are the positions (L). In this form, there are nine components of the hydraulic conductivity in an anisotropic material, which can be placed in matrix form to give what is known as the "hydraulic conductivity tensor" [19]:

$$K = \begin{bmatrix} K_{xx} & K_{xy} & K_{xz} \\ K_{yx} & K_{yy} & K_{yz} \\ K_{zx} & K_{zy} & K_{zz} \end{bmatrix} \tag{4}$$

Assuming that the principal directions of anisotropy coincide with the $x$, $y$, and $z$ directions of the coordinate axes, the six components $K_{xy}$, $K_{xz}$, $K_{yx}$, $K_{yz}$, $K_{zx}$, and $K_{zy}$ are all equal to zero. In this case, Equation (4) is simplified as [19]:

$$K = \begin{bmatrix} K_{xx} & 0 & 0 \\ 0 & K_{yy} & 0 \\ 0 & 0 & K_{zz} \end{bmatrix} \tag{5}$$

In this research, only flow in the $x$ and $y$ directions is considered. Therefore, for homogeneous and anisotropic conditions, Equation (5) is simplified to:

$$K = \begin{bmatrix} K_{xx} & 0 \\ 0 & K_{yy} \end{bmatrix} \tag{6}$$

### 3.2. Determination of the Plane of the Potentiometric Surface and Particle Positions

The seepage velocity ($LT^{-1}$) under homogeneous and isotropic condition is:

$$v = \frac{q}{\varphi} = \frac{-K\frac{dh}{dl}}{\varphi} \tag{7}$$

where, $l$ is the position in the direction of flow (L). $\varphi$ is effective porosity (dimensionless). For homogeneous and isotropic conditions, the hydraulic conductivity was set as $K = 0.0001$ m/s, porosity was $\varphi = 0.25$ [17].

This research uses field data from three or more wells at a time interval ($i$). A regression is performed to obtain a solution for the plane of the potentiometric surface or water table elevation ($h$):

$$h(x, y)_i = A_i x + B_i y + C \tag{8}$$

where, $x$ and $y$ is a position of interest, $A$ is the gradient of head in the $x$ direction (dimensionless), $B$ is the gradient of head in the $y$ direction (dimensionless), $C$ is a constant defined as the elevation of the water table at $(0,0)$ (L), and $i$ is the time interval.

The driving force for the groundwater flow is the hydraulic gradient. Given the plane of the potentiometric surface/water table elevation, gradients in the $x$ and $y$ directions can be resolved for specified time intervals. For homogeneous and isotropic conditions, the positions of particle moving at each time step $i$ are:

$$x_{i=0} = x_{initial}, \text{ and } y_{i=0} = y_{initial} \tag{9}$$

When taking a particle forward in time:

$$x_{i+1} = x_i + \Delta x_i, \text{ and } y_{i+1} = y_i + \Delta y_i \tag{10}$$

where

$$\Delta x_i = v_x \Delta t_i = \frac{-KA_i}{\varphi}\Delta t_i, \text{ and } \Delta y_i = v_y \Delta t_i = \frac{-KB_i}{\varphi}\Delta t_i \tag{11}$$

where, $\Delta t$ = time (T), $v_x$ and $v_y$ is the seepage velocity in the $x$ and $y$ directions, respectively (LT$^{-1}$).

### 3.3. Homogeneous and Anisotropic Conditions with Retardation

The positions of a particle moving under homogeneous and anisotropic conditions at each time step are calculated as follows,

$$x_{i+1} = x_i + \frac{-K_x A_i}{R\varphi}\Delta t_i, \text{ and } y_{i+1} = y_i + \frac{-K_y B_i}{R\varphi}\Delta t_i \tag{12}$$

where, $R$ is the retardation factor, which is calculated as:

$$R = 1 + \frac{K_{oc} f_{oc} \rho_b}{\varphi} \tag{13}$$

where, $K_{oc}$ is the partition coefficient (L$^3$M$^{-1}$) (M represents the mass), $f_{oc}$ is the weight fraction of organic carbon [dimensionless], and bulk density $\rho_b$ =1.987 kg/L [17]. In this case, hydraulic conductivities in the x and y directions are set as $K_x$ = 0.0001 m/s and $K_y$ = $5 \times 10^{-5}$ m/s, respectively [17]. $K_{oc}$ and $f_{oc}$ are set as 63 mL/gm and 0.01, respectively [20]. It is worth noting that in an equivalent system where the velocities of groundwater and contaminant are the same, there is no influence of retardation on the contaminant transport. Therefore, use of $R$ values greater than 1 are only applicable to circumstances where the contaminant is advancing into media that has not been previously contacted by contaminants.

### 3.4. Degradation Kinetics of Contaminants

It is assumed that the degradation of a subsurface contaminant follows the pseudo first-order kinetic reaction. The following equation was employed in the model:

$$C_i = C_0 e^{-k \sum_{i=0}^{n} \Delta t_i} \tag{14}$$

where, $C_i$ is the concentration at the time interval $i$ (ML$^{-3}$), $C_0$ is the initial concentration (ML$^{-3}$), $k$ is rate constant (T$^{-1}$), and $\Delta t_i$ is the $i$th time interval (T). The first order degradation rate constant of RDX $k$ is set as 0.063/day [20]. The minimum concentration of subsurface contaminant at the study site is assumed to be 0.005 mg/L.

## 4. Results

Figure 5 shows the magnitude of the hydraulic gradient in each direction at PCD over the study period. The center of the circle represents the starting point of groundwater or a fluid particle. The red lines in the figure indicate the directions and magnitudes of the hydraulic gradients for groundwater or fluid particles flow from the starting point. The direction of gradient ranges from east-northeast to west-southwest. Specifically, directions of hydraulic gradient are diversified in half of the chart

with different magnitudes, and the higher magnitudes (0.005–0.01) are mainly distributed along the northeast to southwest direction. During brief periods, the direction of the hydraulic gradient shifts to the northwest.

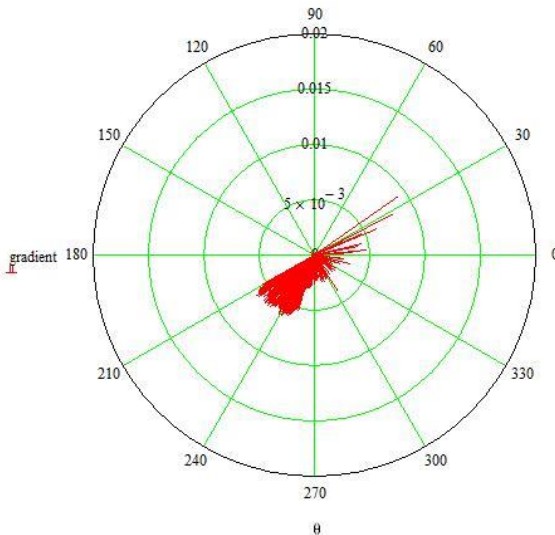

**Figure 5.** Rose chart for hydraulic gradients vs. angles at PCD, CO.

The results of particle tracking pathlines under homogeneous and isotropic conditions at PCD, Colorado, are shown in Figure 6. The three minimum water levels shown in the figure are marked by cycles in the hydrograph. High groundwater levels mainly occurred in the wet season (e.g., summer) from 2006–2008. However, dry seasons which took place in the winter resulted in the change in groundwater flow direction, which are correspond to the three flow direction reversals in the pathline (Figure 6a). For example, from June 2006 to September 2006, the dry season drove the water level to decline, leading the groundwater flow directions changed during the period (blue circle in Figure 6a). From then on the precipitation in wet season forced the water level to rise, which caused the particles to flow back to the original direction. Similar behavior can be observed for the periods from July 2007 to October 2007 (purple circle) and from July 2008 to September 2008 (orange circle), in which the changes in water level were both driven by the variations in hydrologic conditions.

Hydraulic gradient is the driving force for groundwater flow, therefore, direction of the flow should be the same with that of hydraulic gradient shown in the rose chart (Figure 5). However, as shown in Figure 6a, although it has three direction reversals, groundwater flow mainly centers in one direction over the study period: from northeast to southwest. A proper explanation for this phenomenon is that the hydraulic gradients with higher magnitude are mainly distributed along the northeast to southwest direction, while the small ones are ranged in other directions (Figure 5). According to Darcy's law, hydraulic gradient with higher magnitude may drive particles a longer distance compared to that of smaller gradient. Thus, the higher hydraulic gradient along the northeast to southwest direction dominated the groundwater flow in the period.

Except for the period with missing data (e.g., 14 November 2007 to 22 April 2008), in most of the study period water level was maintained at a relatively high stage and there were just three considerable fluctuations (Figure 3). This is the reason why the hydraulic gradient did not range in various directions (Figure 5), which made the flow direction stay in one major way: from the northeast to the southwest (Figure 6a). Correspondingly, the gradients along the northwest to southeast direction, which take up a small portion, were responsible for the changes in flow direction in the dry season.

By assuming subsurface medium is homogeneous and isotropic, and the source area is located in the downside of the monitoring well ESTCP 11 as shown in in Figure 6b, flow pathline of particles can be tracked in the study site (Figure 6b). Similarly, flow pathlines in homogeneous, anisotropic conditions with retardation and homogeneous, anisotropic conditions with reaction conditions can also

be obtained in the study site using the geological parameters in this study. Therefore, the analytical method developed in this study could be used to track particles given specific geologic parameters. As mentioned before, the focus of this study is to develop an analytical method to track particles. Although geological and geochemical conditions of the study site are referred to the local reports, specific and detailed local geologic parameters need to be investigated before using this analytical solution.

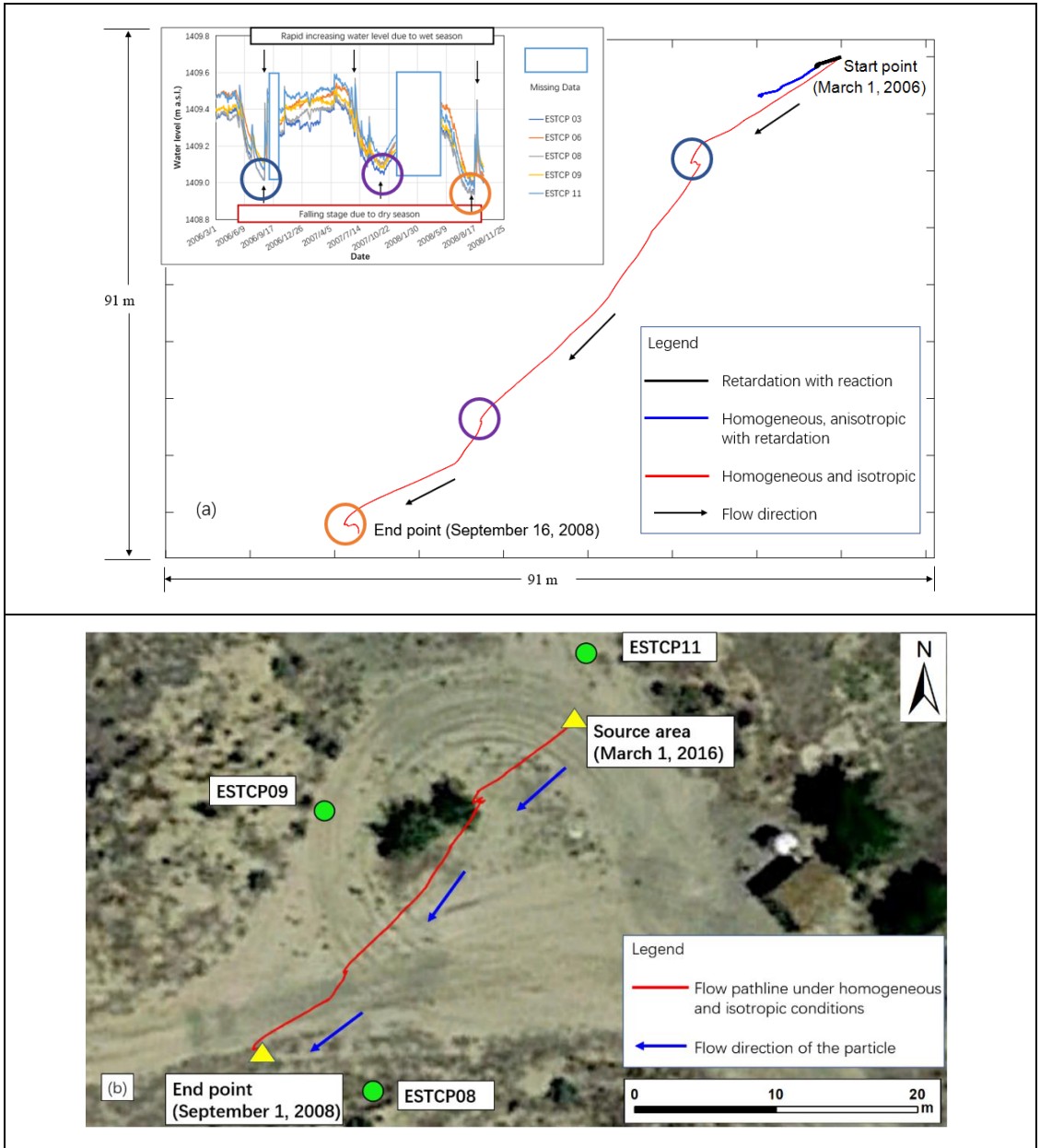

**Figure 6.** (**a**) Flow pathline of particle tracked under in different geologic conditions at Pueblo chemical depot in Pueblo, CO (circles with specific color correspond to flow direction reversals). (**b**) Flow pathline of particle tracked under homogeneous and isotropic conditions by assuming a source area at the study site.

Under homogeneous and isotropic conditions, particle tracking can also be used to back track the source of particles or contaminants. For backward tracking, particle flow direction in each time step should be the same with that of forward tracking. Therefore, the pattern of particle flow pathline for

backward tracking is the same with that of forward tracking and the starting position of the particle in the forward tracking is the ending position of the backward tracking (Figure 6).

Particle tracking pathline under homogeneous and anisotropic conditions with retardation is shown in Figure 6a (blue line). Particle flow direction is almost the same as the scenario under homogeneous and isotropic conditions. However, compared to the scenario without retardation, less distance is moved by the particle due to absorption effect by subsurface medium in this scenario (see in Table 1). In particular, because the hydraulic conductivity in the $x$ direction is assumed to be higher than that in the $y$ direction, particles tended to flow in the $x$ direction such that the flow pattern is flatter in comparison to the isotropic condition.

**Table 1.** The distance particle moved in each scenario at the Pueblo chemical depot in Pueblo, CO.

| Anisotropy | Direction | Retardation | Degradation | $\Delta x$ (Particle Moves in the $x$ Direction) (m) | $\Delta y$ (Particle Moves in the $y$ Direction) (m) | Total Distance (m) |
|---|---|---|---|---|---|---|
| No | Forward | No | No | 17.9 | 25.4 | 31.1 |
| No | Backward | No | No | 17.9 | 25.4 | 31.1 |
| Yes | Forward | Yes | No | 9.8 | 6.9 | 12.0 |
| Yes | Forward | Yes | Yes | 2.9 | 1.8 | 3.4 |

Degradation of organic contaminants in groundwater can occur naturally, supported by available electron donors, electron acceptors and nutrients, or through human intervention using enhanced or engineered bioremediation technologies [21]. The concentration of RDX in groundwater is limited as less than 0.005 mg/L [22]. A flow pathline of the RDX from its sources to its standard of limited concentration may provide an important indication of the remediation strategies, which is critical for the protection of groundwater resources therein. It is assumed that the initial RDX concentration at the site was 1000 mg/L, based on the rate constant of the first-order kinetic reaction of RDX (Section 2), the flow pathline of the contaminant before it reaches the standard of limited concentration (0.005 mg/L) is shown in Figure 6a (black line). The results show that the concentration of RDX was degraded from 1000 mg/L on 1 March 2006 to 0.005 mg/L on 25 October 2006 and the distance the particle moved is much shorter than the scenarios without considering natural attenuation (Table 1 and Figure 6).

## 5. Discussion

The flow pathlines of particles at the study site are dependent on local temporally varying hydrologic conditions. Based on the results, flow directions of particles were changed during the rise and decline of water level. For particle tracking at PCD, CO, the main factors making water level change are seasonal variability, for example, precipitation and evapotranspiration during wet and dry seasons. Although directions of hydraulic gradients were diversified at different time steps, particles are mainly driven by hydraulic gradients with higher magnitudes.

Therefore, the changes of water level caused by precipitation and evapotranspiration at the field site can give some clues about the direction of groundwater flow or contaminant transport. In particular, the large changes of water level caused by the seasonal variations of precipitation and transpiration may influence the main direction of groundwater flow. Furthermore, for the assumption of an aquifer with homogeneous media, the method employed in this study based on dynamic water levels and chemical and reaction properties of the contaminant of interest also provides an indication on contaminant sources therein. Flow pathlines of contaminant can be used to determine whether it could pose a threat to the surrounding groundwater resources.

## 6. Conclusions

This study employed dynamic water-level data in multiple wells to solve particle tracking at three field sites. To determine the particle flow pathlines, firstly, multiple wells are employed to measure

water levels which are used to determine the plain of the potentiometric surface or water table at each time step. It is reasonable to consider that a fluid particle or a sediment exists on this plain. Further, using the slope of the water table and Darcy's law, particle position at each time step and flow pathlines can be obtained. Contaminants flow in the saturated zone can be tracked not only for the study site described in this study, but if (1) hydrogeological conditions are well known, (2) three or more wells in a field site are drilled, and (3) water level data from each well at each time step are collected.

Based on simple assumptions, this study provides a simple method to track contaminants and groundwater in the saturated zones. The focus of this study is to develop an analytical solution to track particles. Geologic conditions are varied from site to site. As long as the flow mechanisms in geologic mediums is laminar and can be governed by Darcy' law, the analytical solution developed in this study could be applied to track particles in these mediums. It is suggested that engineers or researchers use the method in this study may refer to the detailed geologic conditions or conduct experiments to quantify the parameters. Therefore, future work may make the methods and results of particle tracking more realistic. Firstly, to make particle tracking results more efficient, water-level data could be acquired via wireless connections for real-time monitoring. Furthermore, more exact and complex geologic and biogeochemical conditions and assumptions can be considered both in the saturated and unsaturated zones.

**Funding:** This research received no external funding.

**Acknowledgments:** The author would like to thank Thomas Sale for providing data for this effort.

**Conflicts of Interest:** The authors declare no conflict of interest.

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
