# Peer review of "Particle Tracking Using Dynamic Water-Level Data"

_water, doi:10.3390/w12072063_

Round 1

Reviewer 1 Report

The revised manuscript looks well considered on the parts pointed in the first draft.

Please check on the following parts;

  • Line 146; You mentioned about “rb”. Where was it employed?
  • Line 195; northwest >>> northeast ?

Author Response

Response to Reviewer 1 Comments

Point 1: Line 146: You mentioned about “rb”. Where was it employed?

Response 1: Thank you for the comment. Bulk density  is not employed in equation (7) but equation (13), which is  in Line 170.  Therefore, I moved  from Line 146 to Line 170.  The manuscript has also been revised in Line 146 and Lines 171-172:

Line 146: “the hydraulic conductivity was set as  0.0001 m/s, porosity was 0.25 [17].”

Line 171-172: “where,  is the partition coefficient [L3M-1] (M represents the mass),  is the weight fraction of organic carbon [dimensionless], and bulk density 1.987 kg/L [17].”

Point 2: northwest >>> northeast ?

Response 2: Thank you for the comment. For the hydraulic gradients with higher magnitudes (0.005-0.01), they are mainly distributed along the northeast to southwest direction in the rose chart.  For the hydraulic gradients with small magnitudes, they are distributed along other directions including the direction toward northwest.  So in Line 195, the direction I used is along the northeast to southwest direction.  The relevant descriptions are shown in the Lines 192-195:

“The direction of gradient ranges from east-northeast to west-southwest.  Specifically, directions of hydraulic gradient are diversified in half of the chart with different magnitudes, and the higher magnitudes (0.005-0.01) are mainly distributed along the northeast to southwest direction.”

Reviewer 2 Report

The author has satisfactorily responded to all ealier questions and made the necessary changes to the manuscript. The revised version of the manuscript appears to be good. The introduction giving a clearer explanation of the dynamic water level and cyclic water flow. Figure 1 is now more readable. What is more, description of hydrogeologic cross section has been added in the manuscript. The discussion and conclusions are well conducted. It looks ready for publication as far as I can tell.

Author Response

Response to Reviewer 2 Comments

The author has satisfactorily responded to all earlier questions and made the necessary changes to the manuscript. The revised version of the manuscript appears to be good. The introduction giving a clearer explanation of the dynamic water level and cyclic water flow. Figure 1 is now more readable. What is more, description of hydrogeologic cross section has been added in the manuscript. The discussion and conclusions are well conducted. It looks ready for publication as far as I can tell.

Thank you very much for your comments and suggestions.  They are very helpful for improving my manuscript.

This manuscript is a resubmission of an earlier submission. The following is a list of the peer review reports and author responses from that submission.

Round 1

Reviewer 1 Report

The paper “Particle Tracking Using Dynamic Water Level Data” by Gao proposes a new method to track particles in a contaminated groundwater site. The topic itself could be of interest considering that particle tracking methods could be used to find a contaminant origin and fate of the contaminant in the future; however, this paper has serious flaws in many aspects and thus cannot be accepted to Water in its current form. Some general and specific comments are provided below.

General comments:

1) The author appears to propose a novel method for particle tracking; however, the method seems to be simple time-series data analysis and thus does not provide a new scientific contribution.

2) The paper was not targeted to present the detailed methodology for the proposed method nor to interpret the results of the study site. It is not well focused.

3) The computational methods assume very simple cases without having concrete rationale for that. Also, the methods are only limited to simple cases; thus, flexibility of the method is very limited.

4) The parameters used in the computation do not have rationale. They does not have relevancy to the study site.

5) The key result for the computation is not clearly presented (i.e., Figure 5). It is not even clear to what direction particles move over the study period.

Specific comments:

Line 11: “a novel method”: The essence of the proposed method in this study is not well described in the introduction. For example, whether it is a numerical method, analytical method, or simply time-series data analysis.

Line 43: A backward particle-tracking methods à A backward particle-tracking method

Line 60-62: “The objective of the study is to use continuous water level data to resolve groundwater flow under dynamic conditions.”: In the previous paragraphs in the introduction, the purpose of the study appears to be developing a novel particle tracking method; however, it seems to end up with simply interpreting the water level data in the study site.

Line 62-65: “Three geologic conditions where used for conducting particle tracking: 1) homogeneous, isotropic conditions, 2) homogeneous, anisotropic conditions with retardation, and 3) homogeneous, anisotropic conditions with degradation of contaminants in the subsurface following the first-order kinetics.”: There is no rationale for using these three simple scenarios. Also, considering only three scenarios provide very limited flexibility for the proposed method.

Line 66-69: Each part is not properly numbered. For example, Computational methods are in Section 2, Results are in Section 3, and Discussions are in Section 4. Also, there is no Conclusions section in this paper.

Line 77: Hexahydro-1,3,5-trinitro-1,3,5-triazine (RDX) à hexahydro-1,3,5-trinitro-1,3,5-triazine (RDX)

Line 78 & rest of the manuscript: “8 to 10 feet” à Please use SI unit.

Line 132-133: “For homogeneous and isotropic conditions, the hydraulic conductivity was set as ?= 0.00033 ft/sec, porosity was ?= 0.25, and bulk density ??= 1.987 kg/L.”: Any rationale for this?

Line 157-159: “In this case, hydraulic conductivities in the x and y directions are set as ??= 0.00033 ft/sec and ??= 0.000164 ft/sec, respectively. ??? and foc are set as 63 ml/gm and 0.01, respectively [17].”: Are these parameters relevant to the study site?

Line 168-169: Superscripts for [M/L3], [M/L3], and [T-1].

Line 170-171: “? is set as 0.063/day. The minimum concentration of subsurface contaminant at the study site is assumed to be 0.005 mg/L.”: Any rationale for this?

Figure 5: Please provide the legend for the arrows with each color. Basically, it is not clearly visible to which direction particles are moving during the specific period.

Reviewer 2 Report

The paper describes the surface variation of groundwater using the tracking the movement of fluid particles including chemical contaminants. Long term observation using three wells were carried out in a rural chemical industry site in Colorado. The observed results demonstrated that the flow directions were changed during the rise and decline of water level.

The analysis method in the research is unique and it is useful to catch the seasonal characteristic of groundwater surface. The publication of the paper becomes essential to how to track the fluid particle on the groundwater and to study the long-term water surface data. However, the recommendation and Conclusion is not included in the sentences.

The other revision parts are as follows;

  1. Line 79-80; Sand layer is not included in the surface of groundwater. What type of material is covered the surface of ground water?
  2. Line 95   ; Maybe 3. Methods
  3. Line 106, Figure 3.; Please indicate the axis x, y and z.
  4. Line 115; Please make the definition of L, T and M
  5. Line 172;   Maybe 4. Results
  6. Line 178, Figure 4.; If possible, please indicate the upstream and downstream of each vector.
  7. Line 257; Please insert ‘Conclusions’

Reviewer 3 Report

Introduction:

The introduction is well conducted. It addresses the issue and refers to relevant literature. However, please emphasize on works recently done (not older than 5 years). Please provide more examples (papers in journals, also in Water). In my opinion it is important to focus on dynamic water level and cyclic water flow. Similar tests were conducted by Miszkowska A., Lenart S., Koda E. (2017). Changes of permeability of nonwoven geotextiles due to clogging and cyclic water flow in laboratory conditions, Water, 9 (9), 660.

Materials and methods:

This chapter is well conducted.

Figure 1a should be improved, it is not clear.

L78-79. Please add the Hydrogeological cross section of the study area.

L96-100. Darcy's law is valid for laminar flow through sediments. In fine-grained sediments, the dimensions of interstices are small and thus flow is laminar. Coarse-grained sediments also behave similarly but in very coarse-grained sediments the flow may be turbulent. Hence Darcy's law is not always valid in such sediments. Please comments that statement and explain why the linear flow was accepted.

Results:

L234. Please explain why the concentration of RDX was degraded from 1000 mg/L to 0.005 mg/L.

Discussion; Conclusions:

The discussion should be exstended and completed. Please add more works.

Probably section „discussion” should be named „conclusions”?

Please look at this statement:

L68. „Conclusions and recommendations for future work are finally presented in Section 5”. This section is missed! The paper should be put in order.